# Antigen presentation kinetics control T cell/dendritic cell interactions and follicular helper T cell generation in vivo

Robert A Benson[1], Megan KL MacLeod[1], Benjamin G Hale[2], Agapitos Patakas[1], Paul Garside[1], James M Brewer[1]*

[1]Institute of Infection, Immunity and Inflammation, College of Medical, Veterinary and Life Sciences, University of Glasgow, Glasgow, United Kingdom; [2]Institute of Medical Virology, University of Zurich, Zurich, Switzerland

**Abstract** The production of high affinity, class switched antibodies produced by B cells hinges on the effective differentiation of T follicular helper (Tfh) cells. Here we define conditions specifically enhancing Tfh differentiation and providing protection in a model of influenza infection. Tfh responses were associated with prolonged antigen presentation by dendritic cells (DCs), which maintained T cell/DC interactions into stage 3 (>72 hr) of activation. Blocking stage 3 interactions ablated Tfh generation, demonstrating a causal link between T cell-DC behaviour and functional outcomes. The current data therefore explain how duration of antigen presentation affects the dynamics of T cell-DC interactions and consequently determine Tfh cell differentiation in the developing immune response.

## Introduction

Following stimulation via the TcR, selective differentiation of T helper cell subsets is dependent on transcription factors, with Bcl-6 being responsible for T follicular helper (Tfh) cell differentiation (*Johnston et al., 2009*; *Choi et al., 2011*) and antagonism of other T cell phenotypes (Th1, Th2, Th17). Gene knockout studies in mice have revealed the requirement for CD28 mediated-costimulation in the upstream control of Tfh differentiation (*Linterman et al., 2009*), with subsequent signalling through inducible T cell costimulator (ICOS) involved in consolidation of differentiation (*Crotty, 2011*; *Xu et al., 2013*). The mechanisms controlling the induction of this differentiation program are less clear, a role for the cytokines IL-6 and IL-21 and subsequent signalling via STAT3 have been implicated (*Nurieva et al., 2008*), while other studies indicate that the strength and duration of the T cell receptor (TCR) signal itself is the major factor determining effector cell differentiation to Tfh (*Fazilleau et al., 2009*; *Tubo et al., 2013*). Using well defined and controlled conditions that preferentially induce Tfh responses in vivo, we have therefore adopted a reductionist approach to determine which of these factors are important in the differentiation of this subset in vivo.

A variety of particle formulations, including alum, emulsions and liposomes have received sustained interest as vaccine adjuvants for over 80 years (*Brewer, 2006*; *Xiang et al., 2006*; *De Temmerman et al., 2011*). Their ability to affect the availability of antigen and antigen presentation has frequently been linked to their impact on the development of T cell responses (*Constant and Bottomly, 1997*), but causal evidence has been absent. More recently, studies have suggested that the use of nanoparticles as adjuvants can selectively induce Tfh responses (*Moon et al., 2012*). Clearly selective differentiation of Tfh effector cells has implications in vaccine design and pathogen control as for the majority of vaccines the principal effector mechanism induced is production of high affinity, class

*For correspondence: James.
Brewer@glasgow.ac.uk

Competing interests: The
authors declare that no
competing interests exist.

Reviewing editor: Shimon
Sakaguchi, Osaka University,
Japan

**eLife digest** The immune system protects the body from infections, cancer and other diseases. Invading microbes and cancerous cells exhibit proteins that are not normally found in the healthy cells of the body. Fragments of these molecules—known as antigens—may be detected by the immune system, which can then respond by producing antibodies and other responses that try to destroy the threat.

Antibodies are produced by one type of immune cell (known as B cells) with the help of other cells called follicular helper T (Tfh) cells. During an immune response, Tfh cells form from 'naïve' T cells that have not encountered an antigen before. This process has several stages and is activated when the naïve T cells interact with antigens that are displayed on the surface of dendritic cells and other immune cells. However, it is not clear exactly how this process works.

Here, Benson et al. studied the formation of Tfh cells in mice in response to antigens of different sizes. The experiments show that the dendritic cells displayed larger antigens for longer periods of time than they displayed the smaller antigens. Both the small and large antigens allowed dendritic cells to interact with T cells. However, only the dendritic cells that displayed the larger antigens maintained the interaction with T cells for longer periods of time (into the last stage of Tfh cell formation). This enhanced the production of Tfh cells, which boosted the production of antibodies against the antigens to generate immunity to infection.

Further experiments found that blocking the interaction between dendritic cells and T cells during the final stage of Tfh cell formation reduced the production of Tfh cells. Benson et al.'s findings show that the length of time that dendritic cells present antigens on their surface affects the production of Tfh cells and subsequent immune responses. Since Tfh cells are critical to the formation of long-lasting immunity against a virus, these findings could aid efforts to develop more effective vaccines against influenza and other diseases.

switched antibodies by B lymphocytes. Here we employ defined antigen particles to selectively induce Tfh cells and antibody responses and show that increased duration of antigen presentation and subsequent T cell/dendritic cell (DC) interactions are associated with Tfh induction. The functional significance of these late stage (72 hr) interactions in T cell differentiation was demonstrated by blockade of T/DC interactions in vivo. These studies demonstrate that by manipulating the form of antigen delivered late stage antigen presentation by DC and their interaction with T cells can be controlled to drive Tfh differentiation in vivo.

## Results

### Antigen size impacts on protective humoral immunity

In the present study, we employed polystyrene nanoparticles due to their inert nature, defined size, narrow polydispersity, and surface functionalisation allowing covalent linkage of antigen to their surface. Polystyrene nanoparticles do not induce ERK activity or subsequent activation of conventional inflammatory pathways in DC (*Karlson Tde et al., 2013*). Our study focused on polystyrene nanoparticle sizes able to drain freely to the LN (*Manolova et al., 2008*). Carbodiimide-mediated coupling of antigens was highly controlled, such that only particle size and physically dependent parameters (surface area, antigen molecules per particle) varied between immunisation conditions; all other conditions, antigen dose, mass of particles, and antigen density were consistent between 40 nm and 200 nm particle immunisations (*Table 1*). Consequently, immunisations with equivalent quantities of protein were not associated with any confounding difference in total antigenic mass and a single variable, particle size could be tested for its impact on antigen specific immune responses. Initial experiments investigating the effect of particle size on antibody responses revealed that immunisation of mice with Ag covalently linked to 200 nm particles induced significantly higher amounts of specific IgG1 and IgG2c than smaller particles or soluble Ag (*Figure 1A,B*). 1000 nm particles failed to induce an antibody response, potentially due to a failure in uptake by APCs and/or ability to drain to the LN. Subsequent studies thus focussed on the significance and

**Table 1.** Comparison of particle conjugates

| Particle size | 40 nm | 200 nm |
|---|---|---|
| Antigen dose/mouse | 100 µg | 100 µg |
| Antigen molecules/mouse | $1.3 \times 10^{15}$ | $1.3 \times 10^{15}$ |
| Latex dose/mouse | 100 µg | 130 µg |
| Number of particles/mouse | $1.4 \times 10^{12}$ | $2.9 \times 10^{10}$ |
| Antigen molecules/particle | 1000 | 20,000 |
| Surface area/particle | 5024 nm² | 125,600 nm² |
| Antigen density | 1 per 5 nm² | 1 per 6 nm² |

Data shows an example of a typical carbodiimide-mediated conjugation of ovalbumin to either 40 or 200 nm. Various parameters relating to a typical s.c. immunisation with 100 µg of protein are shown.

processes by which the 200 nm particles were able to support this antibody response, in comparison to smaller particles (40 nm) and soluble antigen that did not.

The functional importance of the observed antigen specific antibody response was evaluated in a murine influenza virus infection model where antibodies to the hemagglutinin (HA) component of influenza virus are important to protective immunity (*Murphy et al., 1982*; *Gerhard, 2001*), blocking viral attachment and fusion of the viral and host cell membrane (*Knossow and Skehel, 2006*). Three weeks after immunisation with HA linked to 200 or 40 nm particles, emulsified in complete Freund's adjuvant (CFA) or as soluble protein, mice were challenged intranasally with influenza A virus (H1N1; A/WSN/1933 [WSN]). Protection from (*Figure 1C*). While mice immunised with

infection was determined by weight loss significant weight loss was 200 nm HA-particles were protected from virus induced pathology, significant weight loss was observed in groups immunised with 40 nm HA-particles, similar weight loss to that observed in unimmunised or soluble HA treated mice.

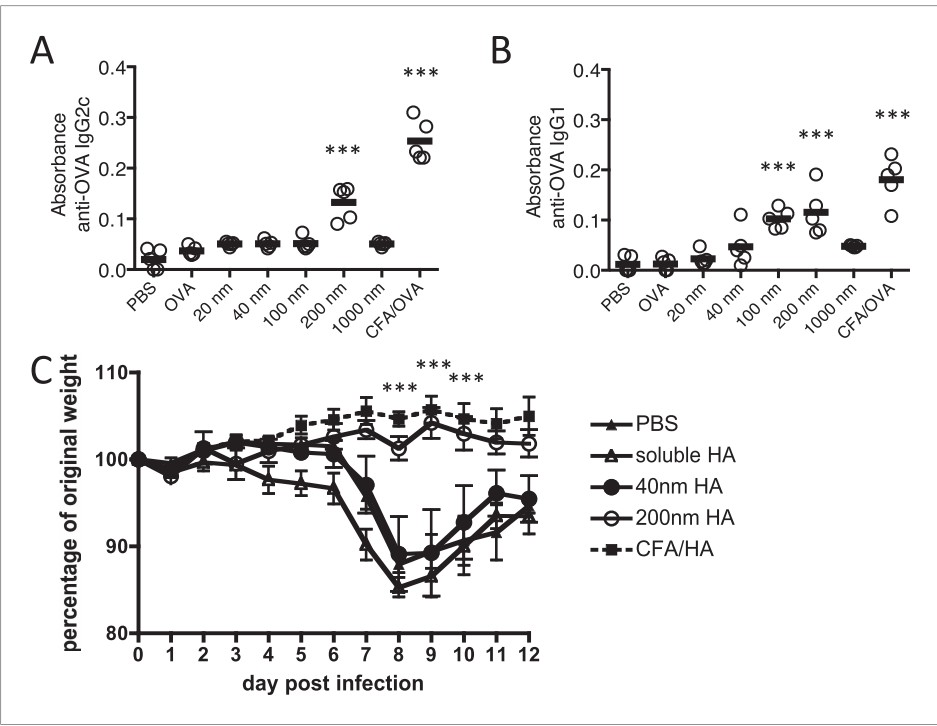

**Figure 1.** Immunisation with antigen loaded 200 nm particles induces elevated IgG and can provide protective immunity to influenza virus infection. Mice were immunised s.c. in the scruff with 100 µg of ovalbumin (OVA) alone or conjugated to 20 nm, 40 nm, 100 nm, 200 nm or 1000 nm particles. 14 days later, serum was tested by ELISA for anti-OVA-IgG2c (**A**) and anti-OVA-IgG1 (**B**). Data show the mean (–) absorbance (405 nm) at a dilution of 1 in 200 with the five experimental animals per group (') ***p < 0.001 comparing to the PBS group. To assess functional responses, C57BL/6 mice were immunised s.c. with 50 µg of hemagglutinin (HA) alone, conjugated to 40 or 200 nm particles, or emulsified with complete Freund's adjuvant (CFA). Weight changes were then monitored following intranasal infection with 300 PFU of influenza A virus (WSN) (**C**). Data shows the mean of five experimental animals per group ± standard deviation ***p < 0.001 comparing 200 nm group to 40 nm group.

## Antigen size influences the development of Tfh and B cell responses

To delineate the mechanisms that allow particle size to control antibody responses, we employed an established adoptive transfer model (*Garside et al., 1998*). Antigen specific CD4[+] T cells and B cells were co-transferred into congenic recipients then immunised with 130 µg of ovalbumin (OVA)-hen egg lysozyme (HEL) conjugate alone, in CFA or bound to 40 nm or 200 nm particles. Analysis of serum samples collected 10 days post immunisation revealed OVA-HEL conjugated to 200 nm particles induced significant levels of serum anti-HEL-IgM[a], whereas antigen conjugated to 40 nm particles did not (*Figure 2A*). Similarly, antigen conjugated to 200 nm particles induced greater expansion of antigen-specific IgM[a+] MD4 B cells than challenge with 40 nm particles 7 days post immunisation (*Figure 2B*). Enhanced expression of germinal centre (GC) markers (GL7 and FAS) following immunisation with 200 nm OVA-HEL particles compared with 40 nm OVA-HEL particles was also evident (*Figure 2C,D*). Conventional GC responses are dependent on activated, antigen-specific CD4[+] T cells (*Owens and Zeine, 1989*; *Nutt and Tarlinton, 2011*), suggesting that

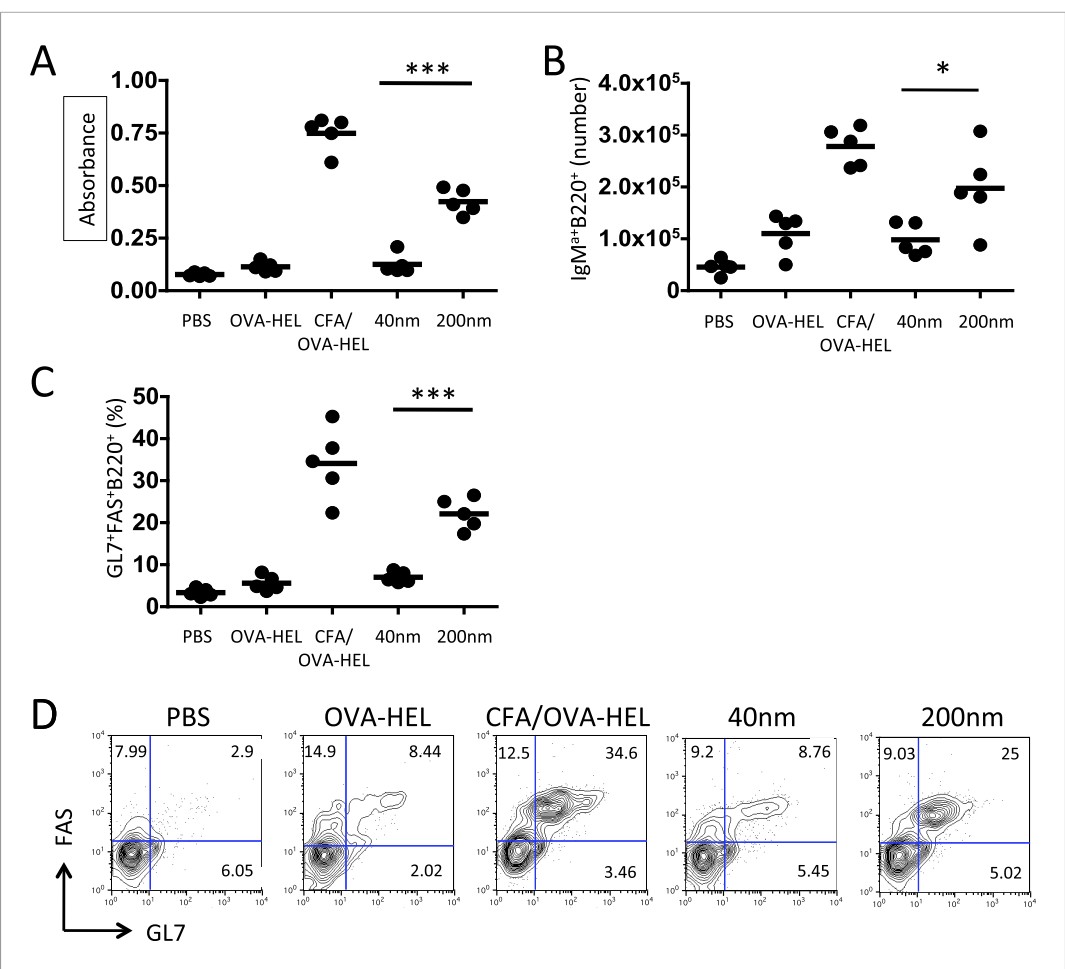

**Figure 2**. Immunisation with antigen loaded 200 nm particles enhances B cell expansion and germinal centre (GC) B cell responses. To dissect the cellular responses in detail, MD4 BcR transgenic (Tg) B cells and OT-II TcR Tg T cells and were adoptively transferred into C57BL/6 recipients. Mice were then challenged with OVA-hen egg lysozyme (HEL) conjugated to 40 nm or 200 nm particles. Challenges with PBS, OVA-HEL alone or in CFA were included as negative and positive controls. Day 10 serum samples were assayed by ELISA to determine anti-HEL-IgM[a] levels (**A**). Expansion of B220[+]IgM[a+] MD4 B cells in popliteal LNs was assessed by flow cytometry 7 days post challenge (**B**). Development of GC B cell responses was determined by examining GL7 and FAS expressing by MD4 B220[+] cells (**C**). Representative staining of GC markers GL7 and FAS are also shown (**D**). Panels show mean value and five individual animals in a representative experiment of 3.

different particles could induce changes in the magnitude or phenotype of the resulting T cell response. However, immunisation with OVA-HEL conjugated to either 40 nm or 200 nm particles could equally increase the proportion and number of antigen specific OT-II T cells in vivo (*Figure 3A,B*), consistent with a previous report (*Gengoux and Leclerc, 1995*). This implied that particle size induced a qualitative rather than quantitative difference in T cell response, favouring generation of the Tfh subset of T cells that have been defined as supporting B cell responses and GC formation (*Ma et al., 2012*). Analysis of Tfh markers CXCR5, PD-1, ICOS and SLAM by adoptively transferred OT-II T cells demonstrated the percentage and number of antigen specific cells differentiating into Tfh cells was very low at day 5 post immunisation with soluble antigen or antigen conjugated to 40 nm particles (*Figure 3C–E*). However, a significant proportion of OT-II

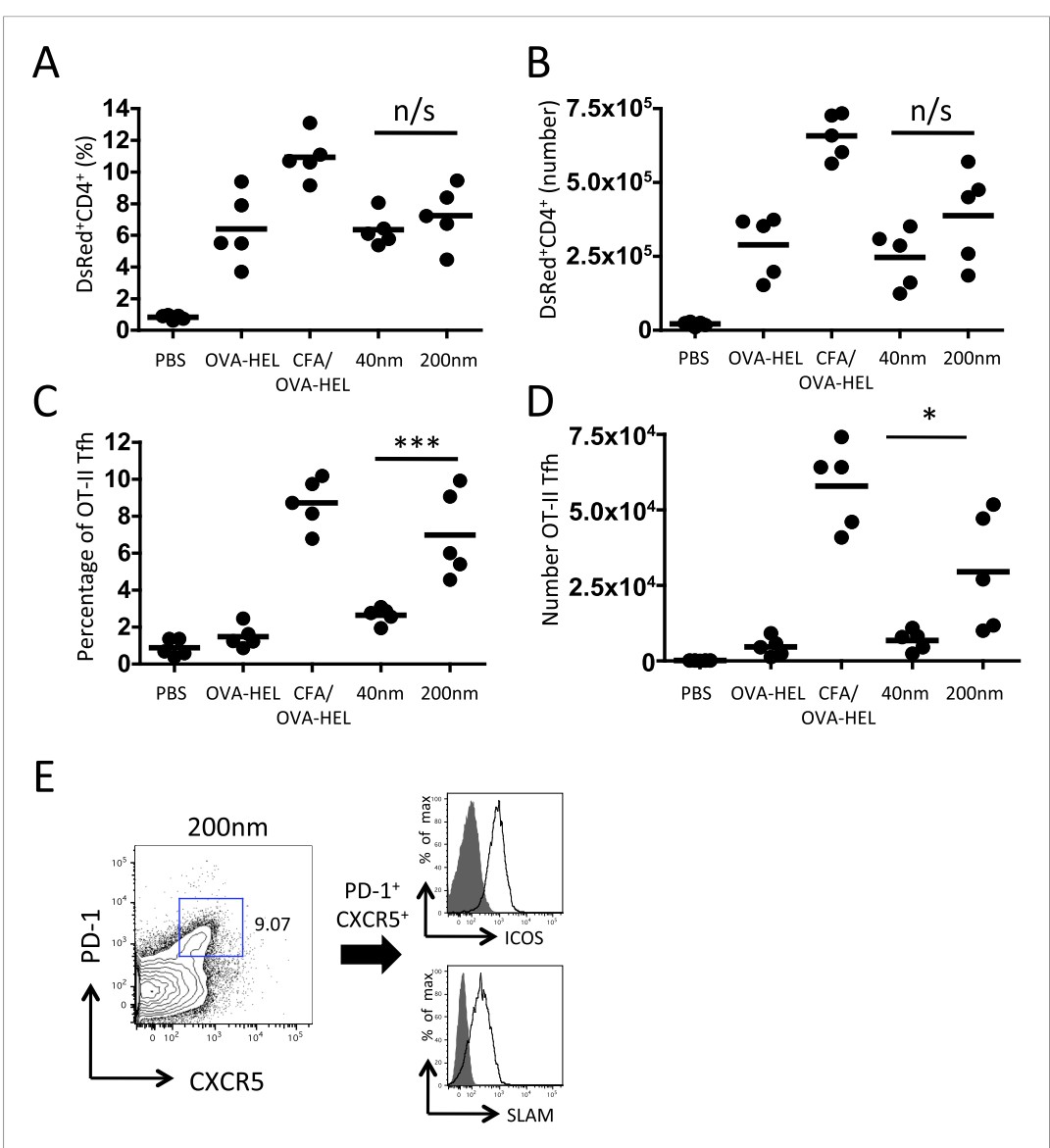

**Figure 3**. 200 nm particles induce T follicular helper (Tfh) differentiation. MD4 BcR Tg B cells and OT-II TcR Tg T cells and were adoptively transferred into C57BL/6 recipients. Mice were then challenged with OVA-HEL conjugated to 40 nm or 200 nm particles. Challenges with PBS, OVA-HEL alone or in CFA were included as negative and positive controls. CD4+DsRed+ OT-II cell expansion in LNs 5 days post challenge (**A**, **B**). These were defined as Tfh cells when being PD-1+ and CXCR5+ (**C**, **D**). Cells were also confirmed as being both ICOS+ and SLAM+ (**E**). Panels show mean value and five individual animals in a representative experiment of 3.

T cells were CXCR5+PD-1+ following challenge with OVA-HEL conjugated to 200 nm particles (*Figure 3C–E*). Changing a single physical characteristic of the immunising antigen selectively enhanced Tfh differentiation without impacting on the magnitude of T cell response, providing an excellent platform to dissect how external stimuli can contribute to Tfh development in vivo.

## Antigen size can determine duration of peptide/MHCII presentation LN APCs

Adjuvants are thought to influence the availability of antigen in vivo and the kinetics of antigen presentation influence consequent CD4+ T cell proliferative, effector and memory responses (*Obst et al., 2005*, *2007*; *Celli et al., 2007*). By covalently linking a traceable antigen, EαGFP (*Pape et al., 2007*), to our particles we were able investigate the impact of particle size on peptide/MHCII presentation in vivo. Immunisation with 40 nm particles resulted in rapid presentation by both DCs and B cells within 6 hr of subcutaneous injection, consistent with published reports (*Itano et al., 2003*; *Pape et al., 2007*). Immunisation with antigen conjugated to 40 nm particles resulted in similar rapid presentation by DCs (*Figure 4A,B*) and B cells (*Figure 5A–C*) as previously observed with soluble antigen (*Itano et al., 2003*; *Pape et al., 2007*). An equivalent proportion of DCs also presented Eα/MHCII following immunisation with 200 nm particles. Percentages and numbers of Eα/MHCII positive DCs equivalent to soluble and 40 nm particle antigen formulations were detected at 24 and 48 hr. However, in contrast to soluble and 40 nm particle formulations, 200 nm particle challenge resulted in presentation of antigen beyond 48 hr, with 21 ± 3% of conventional DCs staining positive for Eα/MHCII at 72 hr post immunisation (*Figure 4C*). Presentation of antigen conjugated to 200 nm particles by the B cell compartment was slower

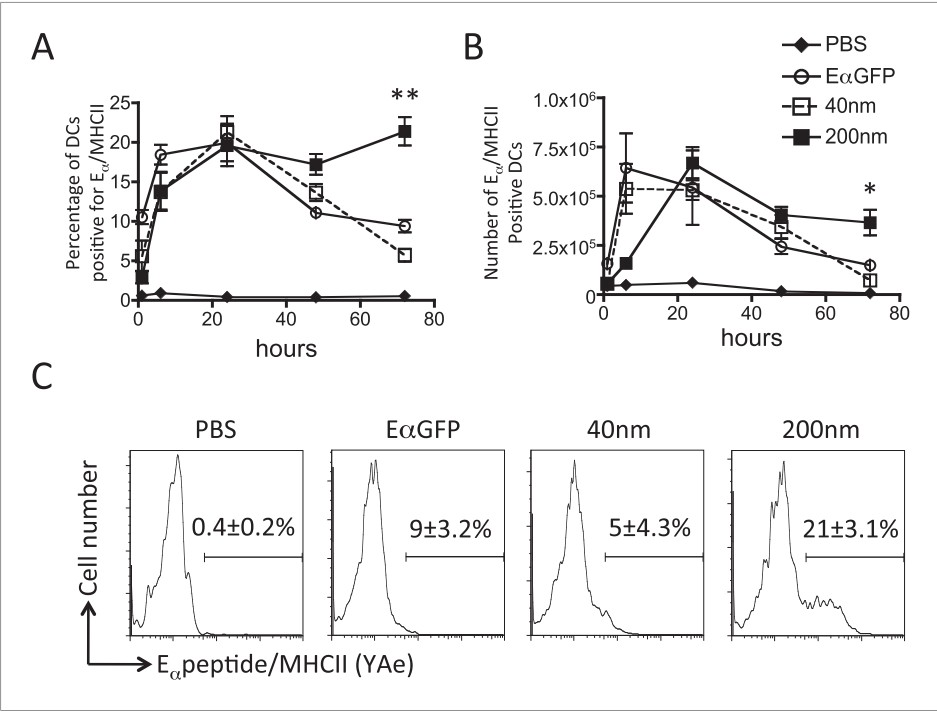

**Figure 4**. 200 nm particulate antigen challenge promotes sustained presentation of peptide/MHCII by dendritic cells (DCs). C57BL/6 mice were immunized s.c. in the footpad with 100 μg of the model antigen EαGFP alone or covalently conjugated to 40 nm or 200 nm particles. Challenge with PBS was included as a negative control. The draining popliteal LNs were harvested after 1, 6, 24, 48 and 72 hr post challenge. Flow cytometry was used to determine levels of presentation of the Eαpeptide/MHC class II complex using the monoclonal antibody YAe on CD11c+B220− conventional DCs. The percentage and number of conventional DCs cells staining YAe+ is shown in (**A**) and (**B**) respectively. Representative flow cytometry plots are shown for 72 hr post challenge (**C**). Data shows the mean of three animals per time point ± standard deviation and are representative of ≥2 independent experiments. * 40 nm vs 200 nm p = 0.012.

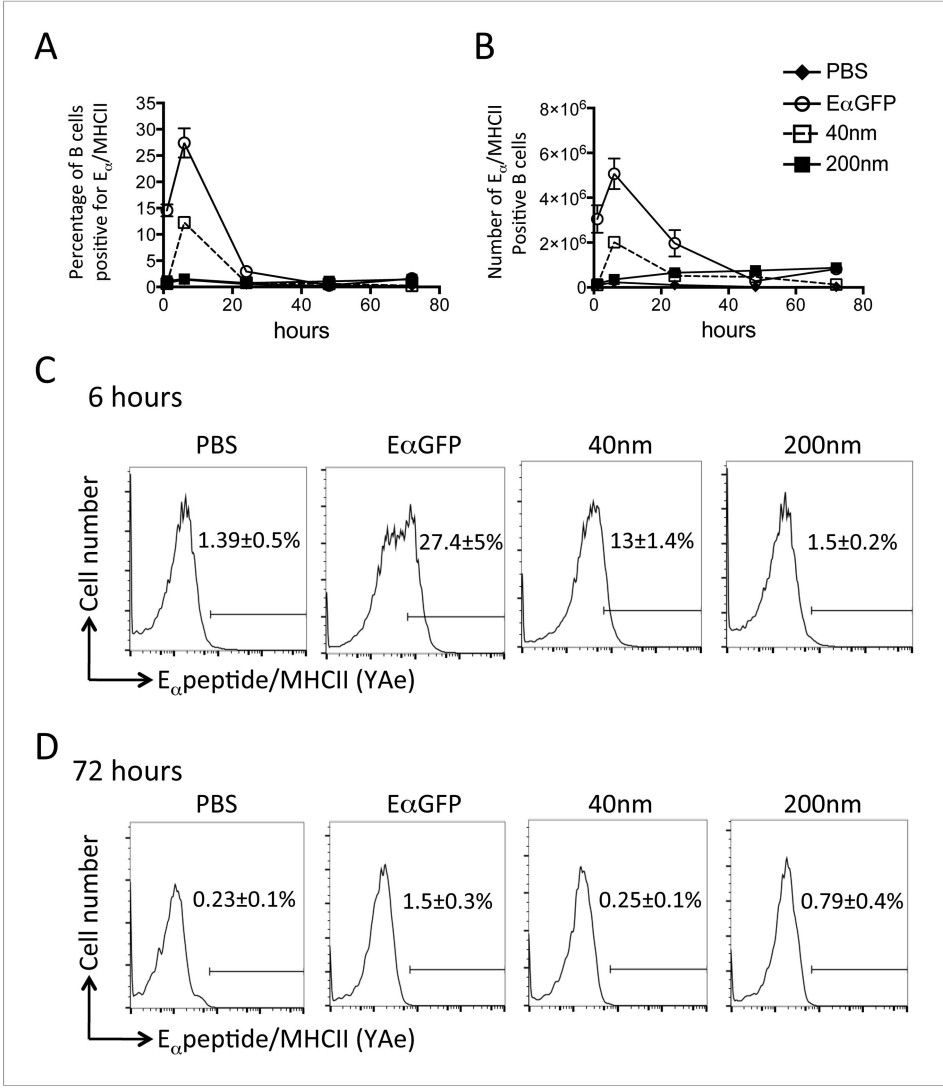

**Figure 5**. Peptide/MHCII presentation by B cells. C57BL/6 mice were immunized s.c. in the footpad with 100 µg of the model antigen EαGFP alone or covalently conjugated to 40 nm or 200 nm particles. Challenge with PBS was included as a negative control. The draining popliteal LNs were harvested after 1, 6, 24, 48 and 72 hr post challenge. Flow cytometry was used to determine levels of presentation of the Eαpeptide/MHC class II complex using the monoclonal antibody YAe on B220+CD11c− B cells. The percentage and number of conventional DCs cells staining YAe+ is shown in (**A**) and (**B**) respectively. Representative flow cytometry plots are shown for 6 and 72 hr post challenge (**C**, **D** respectively). Data shows the mean of three animals per time point ± standard deviation and are representative of ≥2 independent experiments.

(*Figure 5A–C*) but did reach levels equivalent to soluble and 40 nm particle-bound antigen. The sustained presentation observed by DCs was not evident in the B cell compartment (*Figure 5D*), with no significant differences in B cell presentation being detected beyond 24 hr (*Figure 5*). Thus, larger particles did not alter the magnitude of presentation but did prolong the duration of peptide/MHCII presentation by DCs.

## Sustained peptide/MHCII presentation supports late T cell/DC interactions

The relevance of antigen presentation at 72 hr induced by antigen conjugated to 200 nm particles was questionable given evidence from intravital imaging studies. Such studies have demonstrated that T cell priming by DCs occurs in three successive stages (*Stoll et al., 2002*; *Bousso and Robey, 2003*; *Mempel et al., 2004*; *Miller et al., 2004*); stage 1 (0–8 hr) characterised by rapid T cell motility and

short T/DC interactions (2–3 min); stage 2 (12–24 hr) with long-term T/DC interactions associated with the induction of priming and; stage 3 (48–72 hr) where T cells resume motility and short T/DC interactions. We therefore examined the impact of particle size on stage 3 T cell/DC interactions in vivo. CD11c-YFP recipients were adoptively transferred with OT-II DsRed T cells and challenged with OVA conjugated 40 nm or 200 nm particles 72 hr before lymph node imaging by MPLSM (*Video 1* and *Figure 6A*). Multiple short duration T cell-DC interactions (2.149 ± 0.139 min) were observed with 40 nm OVA particles (*Figure 6B*), comparable to that seen with naïve T cells and therefore consistent with an absence of cognate peptide/MHC and classic stage 3 motility (*Mempel et al., 2004*). Notably, interactions longer than 10 min were seen following 200 nm particle challenge (*Figure 6B*), implying that antigen driven cognate recognition was still occurring. This was further supported by the reduced T cell velocity observed in the 200 nm particle group (*Figure 6C*) and again in a lower T cell displacement rate (*Figure 6D*). T cell migratory patterns within the LNs were not significantly different between challenges as evidenced by their equivalent meandering indices (*Figure 6E*). Thus, the antigen presentation by DCs at 72 hr post challenge induced by antigen-conjugated 200 nm particles changed the dynamics of T cell/DC interactions, with stable, long-term interactions extending into the stage 3 time period, conventionally associated with transient interactions and rapid T cell motility (*Hugues et al., 2004*; *Mempel et al., 2004*; *Miller et al., 2004*; *Zinselmeyer et al., 2005*).

## Antigen size influences Tfh differentiation by sustaining peptide/MHCII presentation

Previous studies have demonstrated that differences in stage 2 interactions underlie the outcome of the developing immune response, with long-term (>10 min) and short-term (<3 min) interactions being associated with induction or priming and tolerance respectively (*Hugues et al., 2004*; *Mempel et al., 2004*; *Zinselmeyer et al., 2005*). Significantly, the causal link between T cell behaviour and function was subsequently revealed through disrupting TcR/pMHCII interactions with a monoclonal antibody (Y3P), which resulted in of loss long-term T/DC contacts, a return to rapid T cell motility, and a loss of T cell activation (*Celli et al., 2007*). We hypothesised that the long-term T/DC interactions we observed at 72 hr post immunisation with 200 nm particles were responsible for selective Tfh differentiation. To test this hypothesis, we employed the approach described by *Celli et al (2007)*, using the Y3P monoclonal antibody against MHCII (*Celli et al., 2007*) administered at 72 hr post challenge. Disruption of TcR/pMHCII blocked the stable interactions between T cells and DCs (*Videos 1, 2*, *Figure 7A,B*) and increased T cell motility (data not shown). Unlike blocking stage 2 T/DC interactions (*Celli et al., 2007*), robust T cell proliferative responses were observed with Y3P treatment at 72 hr (*Figure 7C*). However, blocking MHCII at this time point did significantly reduce the proportion of antigen specific cells co-expressing CXCR5 and PD-1 (*Figure 7D,E*). Therefore, we conclude that while the sustained T/DC interactions at 72 hr induced by 200 nm particles do not increase the magnitude of the resulting immune response, these interactions specifically function to alter the quality of immune response by promoting Tfh differentiation.

Analysis of antigen presentation kinetics following immunisation with 200 nm particles revealed that at the time of MHCII blocking (72 hr) the major population of APCs were DCs. Our multiphoton studies further confirmed that T cells were continuing to interact with DCs at this time. However, interactions with antigen presenting B cells could not be ruled out as a contributing factor, especially given their critical role in affirming and maintaining the Tfh phenotype (*Johnston et al., 2009*; *Crotty, 2011*; *Baumjohann et al., 2013*). To establish whether the Tfh phenotype observed following immunisation with 200 nm particles was attributable purely

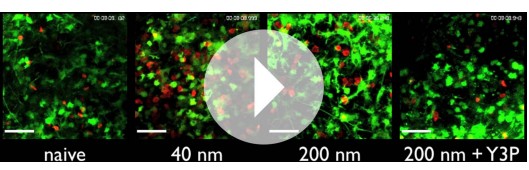

**Video 1.** Imaging DC and T cell behaviour after challenge with 200 nm particulate antigen. DsRed OT-II T cells were adoptively transferred into CD11cYFP recipients and footpad challenged with 100 µg of OVA conjugated to 40 nm or 200 nm particles. Popliteal LNs were imaged at 72 hr. 2 hr prior to imaging, 200 nm challenged groups were given 500 µg mIgG2a or Y3P (anti-mouse I-A). Data is representative of ≥3 individual animals and shows one of three separate areas imaged per lymph node. Scale bar represents 50 µm.

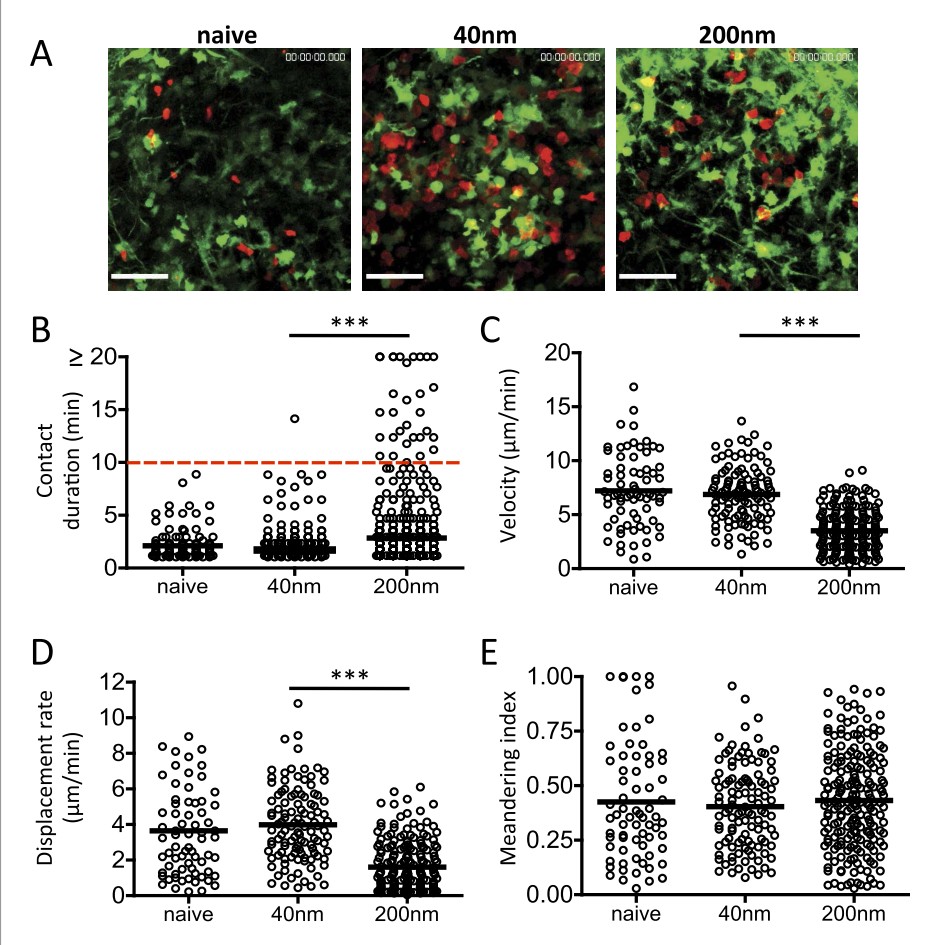

**Figure 6**. DC and T cell interactions persist after challenge with 200 nm particulate antigen. DsRed OT-II T cells were adoptively transferred into CD11cYFP recipients and footpad challenged with 100 µg of OVA alone, conjugated to 40 nm or 200 nm particles. Popliteal LNs were imaged at 72 hr. Representative images following challenge (**A**) correspond with *Video 1* video file. The duration of OT-II T cell interaction with DCs was determined as intersection of DsRed and YFP signals (**B**). The velocity (**C**), displacement rate (**D**) and meandering index (**E**) for T cells were calculated. Data is representative of ≥3 individual animals and shows pooled tracks with mean values of three separate areas of a lymph node. Scale bars represent 50 µm.

to DC presentation at 72 hr we assessed Tfh differentiation in the absence of cognate pMHCII B cell presentation. Immunisation of MD4 mice with 200 nm OVA particles induced equivalent proportional increases in adoptively transferred OT-II T cells as seen in C57BL/6 mice (*Figure 8A*). Interestingly, the Tfh phenotype induced following immunisation with 200 nm particles was not significantly altered in the absence of cognate T-B cell interactions (*Figure 8B*). Inclusion of pMHCII blockade at 72 hr post immunisation in MD4 mice did not influence expansion of the OT-II population (*Figure 8A*) but did significantly reduce the proportion of cells co-expressing CXCR5 and PD-1 (*Figure 8B*), reaffirming the importance of pMHCII recognition at this time. Taken together, the data implicates DCs rather than B cells as the APC population mediating this effect.

## Discussion

We have provided, in unprecedented detail, the spatiotemporal sequence of events that underlie enhanced differentiation of Tfh cells, GC response and protective Ab production by antigen formulations in vivo. By combining highly defined antigen delivery systems, with trackable antigen, antigen-receptor transgenics (Tgs) and state of the art imaging techniques, we revealed that antigen

**Video 2.** Disruption of DC and T cell interactions using Y3P. As for *Video 1*, DsRed OT-II T cells were adoptively transferred into CD11cYFP recipients and footpad challenged with 100 μg of OVA conjugated to 40 nm or 200 nm particles. Popliteal LNs were imaged at 72 hr. 2 hr prior to imaging, 200 nm challenged groups were given 500 μg mIgG2a or Y3P (anti-mouse I-A). Videos were acquired using a 20×/1.0 NA water-immersion objective lens and 2× zoom. Areas of interaction are shown in grey. Scale bars represent 25 μm.

size impacts on the duration of peptide/MHCII presentation and the maintenance beyond 48 hr of functional DC and T cell interactions in the draining LN. The functional relevance of longer DC-T cell interactions, associated with antigen conjugated to 200 nm particles, was dissected by specifically blocking later interactions, resulting in reduced Tfh induction, while the overall magnitude of the T cell response was unaffected. Thus, the temporal characteristics of T cell stimulation can determine their functional differentiation towards a Tfh phenotype, and this can be determined by the size of the particle upon which an antigen is delivered.

Previous studies have investigated the impact of particle size on the immune response to antigen using a variety of formulations, for example lipid vesicles entrapping (*Brewer et al., 2004*; *Moon et al., 2012*) antigens or antigens non-specifically adsorbed to the surface of inert particles (*Mottram et al., 2007*). The inert nature, defined size and surface functionalisation of particles employed in the present study, allowed a single variable, size, to be tested for its impact on antigen immunogenicity. Initial studies simply altering particle size revealed 200 nm particles could induce antibody production following a single immunisation. The functional importance of this observation was startlingly clear, with 200 nm particles able to impart protective anti-HA humoral immunity to influenza infection. Starting with a functional outcome relevant to vaccine design, we sought to dissect the processes by which increasing particle size impacts on the humoral response.

GC formation is central to development of high affinity antibody. GC structures support somatic hypermutation, selection of high affinity B cells and their differentiation into plasma and memory cells (for a comprehensive review see *Victora and Nussenzweig, 2012*). Immunisation with 200 nm particles enhanced this process, explaining our initial observation of increased antibody responses. Essential in this process is the cognate interaction between Ag-specific B and T cells. The nature of this interaction has been the focus of intense research in recent years, culminating in the identification of Tfh cells and the molecules (surface and soluble) involved in their differentiation and function (*Ma et al., 2012*). While both sizes of particle could equally increase antigen specific T cell responses in vivo, we found that larger particles (200 nm) induced greater Tfh differentiation than small (40 nm) particles, consistent with their role in supporting GC responses. Even though the endogenous molecular cues governing the development of Tfh cells are multifactorial (*Crotty, 2011*; *Ma et al., 2012*), understanding how external stimuli can influence T cell differentiation towards this phenotype is less well understood, yet has clear implications in vaccine design. In this case we have demonstrated that simply changing the size of the Ag can clearly impact on Tfh differentiation.

A key event in the initiation of any CD4+ T cell response is the recognition of peptide/MHC class II complexes on DCs. The kinetics of presentation are known to influence the consequent CD4+ T cell proliferative, effector and memory responses induced (*Obst et al., 2005, 2007*; *Celli et al., 2007*). By covalently linking a trackable antigen to the particles we were able to map temporally in vivo peptide/MHC class II presentation and its relation to particle size. Although 200 nm particle immunisation prolonged the duration of peptide/MHC class II presentation by DCs, significantly this did not impact upon the magnitude of the T cell response compared with the smaller 40 nm particle, consistent with a previous report (*Gengoux and Leclerc, 1995*). However, prolonged presentation was associated with the enhanced differentiation of Tfh cells. Antigen availability can dictate the magnitude of a Tfh response, with sustained presentation by B cells maintaining the Tfh phenotype (*Baumjohann et al., 2013*). Notably, immunisation with 200 nm particles resulted in sustained antigen presentation by DCs with cognate interactions with B cells being dispensable for maintenance of a Tfh phenotype. Indeed, sustained antigen presentation associated with Tfh induction has been reported in the absence of cognate interactions with B cells (*Deenick et al., 2010*). In the latter study, CD4+ T cells deficient for SAP, and therefore limited in their ability to interact with B cells, still adopted a Tfh phenotype when a second dose of soluble antigen was given 72 hr later. Presentation at this second challenge was

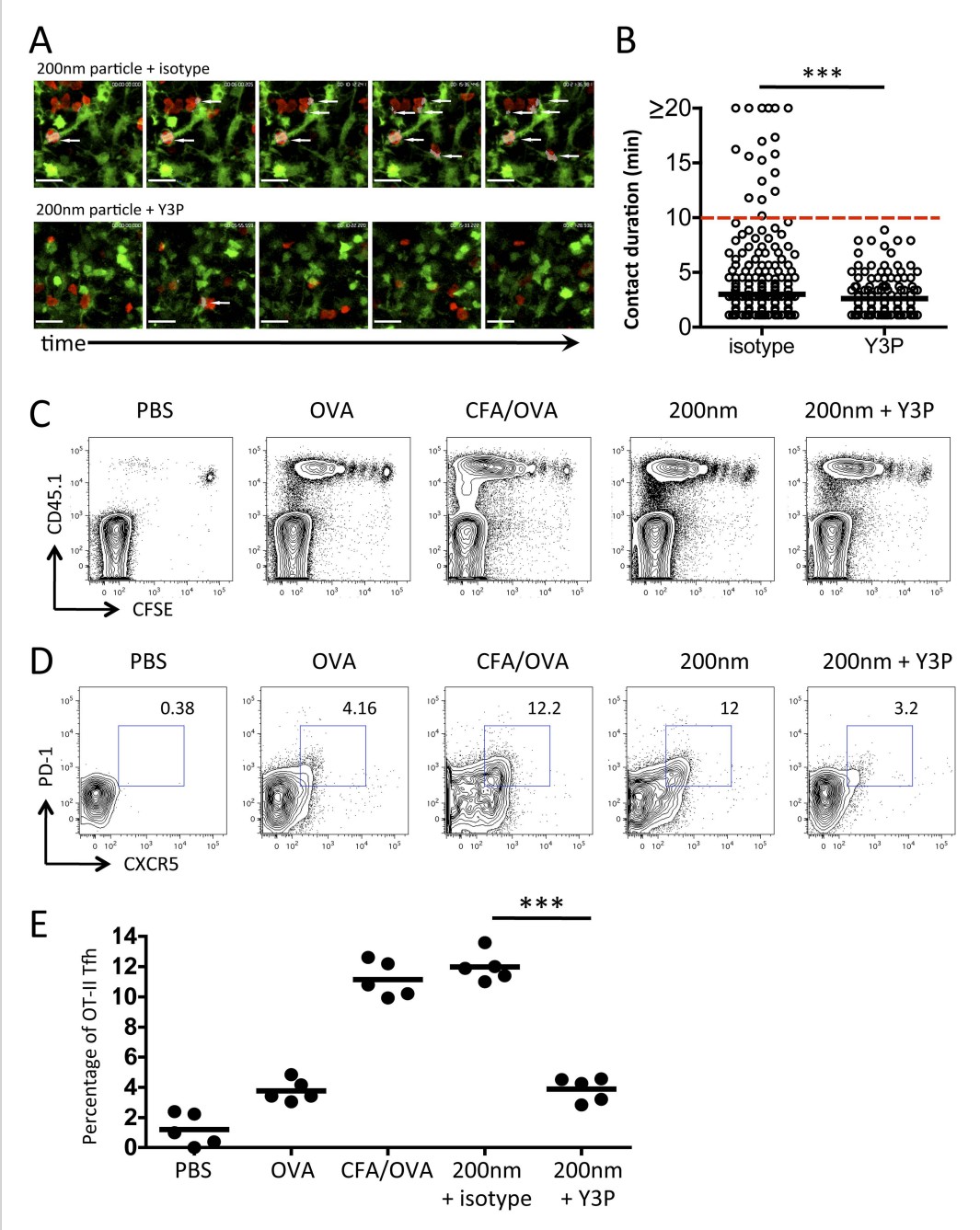

**Figure 7**. Blockade of stage 3 DC/T cell interactions following 200 nm particulate challenge prevents Tfh differentiation. Disruption of stage 3 DC/T cell interaction was demonstrated by imaging DsRed OT-II T cells adoptively transferred into CD11cYFP recipients. Mice were footpad challenged with 100 µg of OVA conjugated to 200 nm particles and popliteal LNs imaged at 72 hr. 500 µg isotype (mIgG2a) or Y3P (anti-mouse I-A) was given i.v. 2 hr prior to imaging. Representative video frames show areas of intersection in grey and highlighted by white arrows (**A**, see **Video 2** for video). Scale bars represent 25 µm. The duration of OT-II T cell interaction with DCs was determined as intersection of DsRed and YFP signals (**B**). Next, CFSE OT-II TcR Tg T cells were adoptively transferred into C57BL/6 recipients. Mice were then challenged with PBS, OVA with CFA or OVA conjugated to 200 nm particles. Mice were treated with a single dose of Y3P or isotype antibody i.v. 72 hr post challenge with particles. Draining LNs were harvested on day 5 post challenge. (**C**) Representative plots showing CFSE division of CD45.1+ OT-II TcR Tg T cells (gated on CD4+ population). Tfh cells were defined as being PD-1+CXCR5+ (**D**) and confirmed as ICOS+ (data not shown) for plotting Tfh proportion as a percentage of the OT-II population (**E**). Data is representative of two independent experiments with five animals per group.

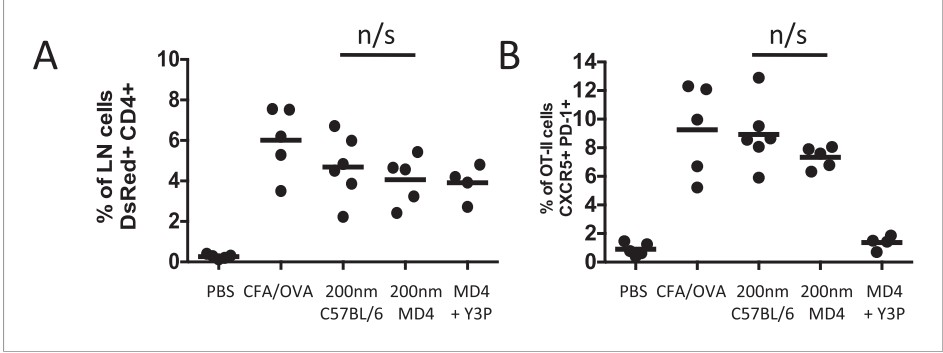

**Figure 8**. Cognate interactions between B and T cells are not required for early Tfh differentiation. OT-II TcR Tg T cells and were adoptively transferred into C57BL/6 or intact MD4 BcR Tg recipients. Mice were then challenged with OVA conjugated to 200 nm particles and 500 µg isotype (mIgG2a) or Y3P (anti-mouse I-A) given i.v. 72 hr later. Challenges with PBS, OVA/CFA were included as negative and positive controls. Flow cytometry was used to assess CD4+CD45.1+ OT-II cell expansion in LNs 5 days post challenge (**A**). Tfh were defined by PD-1 and CXCR5 co-expression (**B**). Panels show mean value and five individual animals, ***p < 0.001.

attributed to DCs (*Deenick et al., 2010*), consistent with our own findings. Indeed, DCs can induce Tfh development in the absence of cognate interactions with B cells (*Choi et al., 2011*) and likely represent the first step initiating Tfh differentiation (*Goenka et al., 2011*). Imaging draining LNs 72 hr post challenge revealed that cognate T cell/DC interactions continued at this time. Importantly the current study adds mechanistic detail to the observational data above; temporal disruption of 72 hr (stage 3) interactions, while not affecting clonal expansion, did inhibit 200 nm particle induced Tfh differentiation. Thus, increasing particle size, and sustaining presentation by DCs and their cognate interactions with T cells underpins the success of the larger particle in driving Tfh differentiation. It is unclear as to how particle size influences temporal availability of pMHCII on DCs. Particulates can impact on the mechanism of antigen uptake and therefore consequent endosomal targeting, processing and dynamics of presentation (*De Temmerman et al., 2011*; *Zhao et al., 2011*; *Ghimire et al., 2012*). Thus the processing of antigen and loading of MHCII and/or its turnover at the cell surface may be contributing factors. In either case, increasing availability of TcR ligand and consequent TcR signal quantity may function in a similar way as increasing TcR dwell time in determining Tfh differentiation (*Tubo et al., 2013*; *Tubo and Jenkins, 2014*).

New delivery systems are required to generate safe and effective vaccines. This has resulted in increased focus on particle delivery systems. Our findings provide mechanistic insight as to why some formulations promote humoral immunity while others do not (*Mottram et al., 2007*; *Moon et al., 2012*). We have linked T cell behaviour to function, demonstrating the importance of stage 3 T/DC interactions in determining the selective differentiation of T cells towards a Tfh lineage. These late stage interactions could be selectively induced using immunisation with antigen conjugated to 200 nm but not 40 nm particles, rationalising approaches to vaccine design using nanoparticles for induction of Tfh, GC formation and high affinity, class switched protective antibody. Previous studies have demonstrated that in the absence of microbial factors, parameters such as the dose and persistence of antigen can affect T cell differentiation (*Constant and Bottomly, 1997*). The current data therefore explain why parameters such as the magnitude and duration of antigen presentation affect subsequent T/DC interactions and consequently determine T cell differentiation in the developing immune response.

## Materials and methods

### Animals

hCD2-DsRed mice (gifted by D Kioussis and A Patel, National Institute for Medical Research, London) were crossed with OVA specific OT-II (*Barnden et al., 1998*) TCR Tg mice. MD4 BcR Tg mice (*Mason et al., 1992*) on C57BL/6 backgrounds were used as a source of HEL specific B cells. 6–8 week old C57BL/6J mice were used as recipients (Harlan, Bicester, UK). CD11c-YFP mice (*Lindquist et al., 2004*)

were used a recipients in multiphoton imaging experiments. All animals were specified pathogen free and maintained under standard animal house conditions at the University of Glasgow in accordance with UK Home Office Regulations.

## Antigen and bead preparations

The recombinant HA protein used was expressed from DNA encoding the extracellular domain of the influenza A virus subtype H1N1 (A/WSN/1933) HA (Life Technologies, Paisley, UK). OVA-HEL conjugates were prepared using gluteraldehyde to couple the chicken OVA (Worthington Biochemical, Lakewood, NJ, USA) and HEL (BBI Enzymes, Gwent, UK) as published (*Garside et al., 1998*). EαGFP was produced in house as detailed previously (*Rush and Brewer, 2010*). HA, OVA, OVA-HEL or EαGFP were covalently bound to Polybead Carboxylate Microspheres (Park Scientific, Northampton, UK). Briefly, carboxylate-modified nanospheres were incubated with 2.5 mg/ml protein (EαGFP, HA, OVA or OVA-HEL) with 1 mM EDAC (1-ethyl-3-(3-dimethylaminopropyl) carbodiimide; Sigma, Dorset, UK) and 25 mM MES (2-N-morpholino ethane sulfonic acid; Sigma) overnight, in line with manufacturers instructions. Conjugation mixes were then spun in an ultracentrifuge at 4°C for 40 min. Protein concentration in the supernatant was measured to allow the amount bound to the microspheres to be calculated. Preparations were stored at 4°C and used within 3 days of preparation.

## Infection with influenza virus

Mice were immunized subcutaneously (s.c.) in the scruff. Infections with influenza virus subtype H1N1 A/WSN/1933 (WSN) were carried out in mice anesthetized with isofluorane and 20 μl of PBS containing 300 PFU of WSN virus administered intranasally (5 mice per group). This dose was used as previous experiments showed that this dose led to approximately 20% weight loss in unvaccinated animals. Viral stocks were prepared and titred in MDCK cells.

## Adoptive transfer of antigen specific lymphocytes and in vivo challenge

Adoptive transfers were performed as described previously (*Smith et al., 2000*). Briefly, single cell suspensions of LNs and spleens were prepared from OT-II and MD4 mice. The proportion of Tg B and T cells was determined by flow cytometry and 2–3 × 10⁶ Tg B cells and 2–3 × 10⁶ Tg T cells adoptively transferred to age-matched C57BL/6J recipients via tail vein injection. For imaging studies, DsRed expressing CD4⁺ OT-II T cells were magnetically sorted from OT-II × hCD2-DsRed animals using mouse CD4⁺ T cell isolation kits (Miltenyi Biotec, Surrey, UK). Mice were immunized via the footpad with 130 μg of OVA-HEL conjugate alone, with CFA (Sigma, Dorset, UK), or covalently bound to carboxylate-modified microspheres. Injection with 50 μl of PBS was used as a vehicle control. Popliteal LNs were excised for ex vivo study. In MHCII blocking experiments, 500 μg Y3P (BioXcell, New Hampshire, USA) or isotype control (mouse IgG2a; BioXcell) was given via tail vein injection. In T cell/DC interaction studies, LNs were imaged 2 hr after antibody treatment.

## Flow cytometry

Single cell suspensions were stained using combinations of anti-CD11c-PE (N418, BD Biosciences, San Jose, CA, USA), anti-B220-PerCP (RA3-6B2, BD Biosciences), anti-I-A$^b$/Ea$_{52-68}$-biotin (YAe, eBioscience, San Diego, CA, USA), anti-CD4-eFluor450 (RM4-5, eBioscience), anti-ICOS-alexa-fluor-488 (C398.4A, BioLegend, San Diego, CA, USA), anti-PD-1-PE-Cy7 (J43, eBiosciences), anti-SLAM-APC (TC15-12F12.2, BioLegend), biotinylated anti-CXCR5 (2G8, BD Biosciences), anti-B220-eFluor450 (RA3-6B2, eBioscience), anti-GL-7-FITC (BD Biosciences), anti-FAS-PE (BD Biosciences) and biotinylated anti-IgM$^a$ (DM-1, BD Biosciences). Biotinylated antibodies were detected by incubation with fluorochrome-conjugated streptavidin (BD Biosciences). Appropriate isotype controls were used throughout. Samples were acquired using a MACSQuant analyzer (Miltenyi Biotec, Surrey, UK) and analysed using FlowJo software (Tree Star Inc, Ashland, OR, USA).

## Multiphoton imaging

Multiphoton imaging was carried out using a Zeiss LSM7 MP system equipped with a 20×/1.0 NA water-immersion objective lens (Zeiss UK, Cambridge, UK) and a tunable Titanium: sapphire solid-state two-photon excitation source (Chameleon Ultra II; Coherent Laser Group, Glasgow, UK) and optical parametric oscillator (OPO; Coherent Laser Group). Excised LNs were continuously bathed in

warmed (37°C), oxygenated $CO_2$ independent medium. A laser output of 820 nm and OPO signal at 1060 nm provided excitation of YFP CD11c$^+$ and DsRed OT-II cells. Videos were acquired for 20 to 30 min with an X-Y pixel resolution of 512 × 512 in 2 μm Z increments. 3D tracking was performed using Volocity 6.1.1 (Perkin Elmer, Cambridge, UK). Values representing the mean velocity, displacement, and meandering index were calculated for each object. The intersection of DsRed and YFP objects was used to determine interaction between T cells and DCs respectively.

## Statistics

Results are expressed as mean ± standard deviation. Gaussian distribution was confirmed by D'Agostino & Pearson omnibus normality test prior to significance being determined by one-way ANOVA. Significance in DC-T cell interaction experiments was confirmed by Mann Whitney. Values of p < 0.05 were regarded as significant.

## Acknowledgements

This work was supported by the Wellcome Trust (Grant Number WT085589MA). BGH is a Sir Henry Dale Fellow jointly funded by the Wellcome Trust and the Royal Society (Grant Number 100034/Z/12/Z). The authors declare no competing financial interests.

## Additional information

### Funding

| Funder | Grant reference | Author |
| --- | --- | --- |
| Wellcome Trust | WT085589MA | Paul Garside, James M Brewer |
| Wellcome Trust | 100034/Z/12/Z | Megan KL MacLeod |

The funder had no role in study design, data collection and interpretation, or the decision to submit the work for publication.

### Author contributions

RAB, Acquisition of data, Analysis and interpretation of data, Drafting or revising the article; MKLML, Analysis and interpretation of data, Drafting or revising the article, Contributed unpublished essential data or reagents; BGH, Conception and design, Acquisition of data, Analysis and interpretation of data; AP, PG, Conception and design, Analysis and interpretation of data, Drafting or revising the article; JMB, Conception and design, Acquisition of data, Analysis and interpretation of data, Drafting or revising the article

### Author ORCIDs

James M Brewer, http://orcid.org/0000-0001-7933-0915

### Ethics

Animal experimentation: All animals were specified pathogen free and maintained under standard animal house conditions at the University of Glasgow in accordance with local and UK Home Office Regulations.

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
