## [Decision Letter]

[Editors’ note: this article was originally rejected after discussions between the reviewers, but the authors were invited to resubmit after an appeal against the decision.]

Thank you for choosing to send your work entitled “Antigen presentation kinetics control T cell/Dendritic Cell interactions and Tfh generation in vivo” for consideration at *eLife*. Your full submission has been evaluated by Tadatsugu Taniguchi (Senior Editor) and three reviewers, one of whom is a member of our Board of Reviewing Editors. Based on discussions amongst the reviewers and the individual comments below, we regret to inform you that your work will not be considered further for publication in *eLife*.

In particular, the reviewers expressed their concern that the paper is short of conceptual novelty and the role of B cells as antigen-presenting cells is not addressed.

Reviewer #1:

T-follicular helper cells have been found to have a critical role in the formation and maintenance of germinal centers and the production of high affinity antibodies. As such, the issue of how best to induce these cells is of considerable importance to vaccine design. In this report, Benson et al. examine the use of polystyrene nanoparticle of different sizes as adjuvants. They find that larger particles lead to formation of more stable peptide MHC complexes on DCs leading to prolonged antigen presentation and longer DC/T-cell contact durations which leads to the enhanced formation of Tfh cells. Consequently antibody production, GC formation and protection from influenza virus infection are all enhanced.

1) This is an interesting study but it almost entirely omits consideration of the critical role of B-cells as APCs in Tfh induction. Previously it has been demonstrated that Tfh form following a two-step process by which T-cells initially interact with DCs which leads to an early wave of pre-Tfh production before a second step of interaction with antigen presenting B-cells that cements the Tfh program (9). It is surprising that the authors have only focused on the first of these two steps and not examined if there is also any effect on B-cell antigen presentation as this would widen and strengthen the article’s conclusions. An effect on B-cells would not be mutually exclusive to the effects on DCs and it seems possible that both would play a role in the increased Tfh formation seen here. The authors correctly state that the Tangye group (11) have demonstrated that it is possible to bypass the requirement for B-cells by adding more antigen and prolonging DC antigen presentation, but even here it remains controversial if DCs alone can induce PD1hi GC-Tfh and high levels of IL-21 expression (16) and where both DCs and B-cells are present B-cells are believed to have a dominant role in Tfh formation from day 3 onwards even in the presence of large amounts of antigen (LCMV infection) (7).

This is important as the authors make clear statements ascribing all the phenomenon to the increased antigen contact in DCs without exploring this alternative possibility. This is particularly acute in Figure 5 where MHCII blocking antibodies are used to interfere with the late stage DC/T-cell contacts. At 72 hours we would also expect B-cell antigen presentation to have an important/dominant role as APCs which would also be blocked by the same antibody, currently it is not possible to say if the effects of the antibody is only by blocking DCs or both DCs and B-cells. It is also worth noting that essentially this point would be testable using the same experimental methods already used i.e. YAe antibody to see if B-cells had altered antigen presentation. Additionally some experiments using vaccination in B-cell deficient mice ideally (μMT, T-cell transfer into RAG-/- or B-cell depletion by anti-CD20) are required to make strong conclusions about the specific role of DCs in the absence of B-cells.

2) A wider range of sizes (e.g. 40, 100, 150, 200nm) would have been useful to more precisely investigate the relationship between particulate size, DC antigen presentation and effectiveness. For example, Mottram et al. used a range of 7 bead sizes and found a bimodal distribution of effectiveness for cytokine induction suggesting that the relationship between size and efficacy may be not always be entirely linear, it would be impossible to make any similar observation in this study since only two sizes are used.

Reviewer #2:

In this paper, Benson et al. use polystyrene nanoparticles coated with various antigens to assess the impact that size has on vaccine responses. They show that coupling the same amount of protein antigen to larger particles increases antibody production, in a similar way to using an adjuvant. This increased influenza vaccine efficacy. They demonstrate that antigen presentation By DCs is prolonged when larger particles are administered, and DCs engage T cells in longer interactions. Finally they demonstrate that blocking antigen presentation at 72hrs post immunisation decreases the proportion of Tfh cells formed. Although it is not clear how the larger particles mediate this effect. Together, this is a nice paper showing that particle size can alter vaccine efficacy, something with direct relevance to human and animal health.

There are a few issues with the manuscript in its current form that should be addressed:

1) In 2011 in JI, Baumjohann et al. described that B cells are dispensable for initial Tfh differentiation (48hrs post vaccination) but played an important role 3.5 days after vaccination for stabilising the Tfh population. There is no discussion in this paper of whether these particles may be acting via Ag-presentation by B cells. Indeed, the evidence for changing the interaction time of T cells and DCs is clear, but the inference that the increased population of Tfh cells observed with large particles is due to these T:DC rather than T:B interactions at 72hrs post immunisation has not been demonstrated. The blocking experiments in Figure 5 would disrupt all antigen presentation, not just that from DCs. A way to determine this experimentally would be to immunise B cell deficient mice (or mice whose B cells don't present cognate antigen) with the different sized particles and see if the expansion of the Tfh population still occurs.

2) In Figure 3 it is clear that the % of DCs presenting antigen is the same for the first 48hrs. Is the amount (or MFI) of antigen presentation similar between particle sizes, particularly at earlier time points?

Benson et al. argue that 200 nm particles prolong the duration of peptide/MHCII presentation on DCs. Does this mean that each individual MHCII molecule is maintained at the cell surface for longer? Or that larger particles deliver processed peptide at a slower rate, perhaps via an alternative route, so that MHCII can be loaded in the ER for longer? Or is it both? Do the authors know how the antigen is processed by the DC when conjugated to 40 nm or 200 nm nanoparticles? Are they processed by different compartments? The stage 3 disruption experiments in Figure 5 define stage 3 simply by time after immunisation; so a slowly loaded MHCII could confound this approach. It would be worth discussing these potential confounders of the interpretation.

Reviewer #3:

In this manuscript, the authors analyze the relative capacity of particulate antigens to optimize antibody response by inducing dendritic cell-mediated Tfh cell development. They found that:

1) Ag conjugated 200 nm particles induced more enhanced antibody responses than 40 nm particles or soluble antigen (Figure 1). These data are redundant with previous literature showing that nanoparticles allow for the reduction of the administered antigen amount compared to immunization with soluble protein and induce strongly enhanced antibody responses (Virology Journal 2013, 10:108).

2) Ag conjugated 200 nm particles induced prolonged presentation of antigen by dendritic cells compared with 40 nm particles or soluble antigen (Figure 3). These data are redundant with previous literature showing that particulate form of antigens are more efficiently processed and presented by different APC, including dendritic cells, macrophages and B cells, than soluble antigen (Journal of Immunology, 1996, 156:2809; Immunology and Cell Biology, 2004 82:506).

3) Prolonged antigen presentation by dendritic cells drove Tfh cell development. These data are redundant with previous literature showing that dendritic cells support prolonged Tfh cell responses when they are forced to sustained antigen stimulation (Journal of immunology, 2011, 187:1091; Immunity, 2010, 33:241).

Although the authors provide a comprehensive examination of the in vivo T/dendritic cell interactions following in vivo administration of particulate forms of antigens, the lack of conceptual novelty tampers down the enthusiasm for this manuscript. In addition, the authors did not evaluate the contribution of antigen presentation by B cells, and this strongly compromises the conclusions of the paper.

Dendritic cell-mediated presentation launches the Tfh cell program. However, later antigen presentation by B cells is important for the maintenance of Tfh cell responses. Therefore, to demonstrate the ability of the dendritic cells to sustain Tfh cell development, the authors should perform experiments in the absence of B cells or in the absence of key cognate T-B interactions.

[Editors’ note: what now follows is the decision letter after the authors submitted for further consideration. Minor issues and corrections have not been included, so there is not an accompanying Author response.]

Thank you for choosing to send your work entitled “Antigen presentation kinetics control T cell/Dendritic Cell interactions and Tfh generation in vivo” for consideration at *eLife*. Your letter of appeal has been considered by Tadatsugu Taniguchi (Senior Editor) and a Reviewing Editor, and we are prepared to consider a revised submission.

Please perform the experiment you outline, using non-cognate B cells is a reasonable way of doing it. In addition, use a wider range of the size of nano particles, as suggested by the reviewers.

---

## [Author Response]

[Editors’ note: the author responses to the first round of peer review follow.]

Thank you for considering our manuscript “Antigen presentation kinetics control T cell/Dendritic Cell interactions and Tfh generation in vivo”. We appreciate the time and effort taken in reviewing it. Understandably, we are disappointed in the decision made by the panel, however we would welcome an opportunity to address the comments made by the referees in a revised manuscript. The two key issues raised by the referees were the lack of novelty and the role of B cell antigen presentation.

Firstly, with regard to the issue of novelty raised by Reviewer 3. While, our use of physically highly characterised particles is not novel, it is controllable, relevant to vaccination and avoids the usual empirical adjuvant approaches widely used in the literature. Our combination of manipulation of T cell/Dendritic cell interactions and the impact on resulting immune response with the analysis of this directly in vivo using real-time imaging is entirely novel. The references quoted by Reviewer 3 are purely descriptive, only demonstrating that particulate antigens are better than soluble at driving an antibody response, an approach that has been used in the vaccine field for some considerable time with no clear mechanism of action. The further work cited does not attempt to delineate the cellular mechanisms underlying how particulate antigens drive Tfh generation and hence antibody production – the novel component of our study. Thus, our manuscript is the first causal in vivo demonstration of how T cell/dendritic cell interactions can be modulated to determine T cell phenotype in a way directly relevant to human and animal health (as stated by Reviewer 2).

With regard to the second point, we already have data requested by Reviewer 2 relating to antigen presentation by B cells. These results show no difference in presentation following immunisation with different particles, which is why we focused the current manuscript on DC. However, if you feel it would strengthen the manuscript, we could include the MD4 studies suggested by the referees, though we believe this would be negative data.

We are currently in a situation where we still lack a mechanistic basis for the function of vaccine adjuvants in vivo despite their constant use in vaccination for over 90 years. Particles form an important part of these vaccine formulations, and we believe that the current manuscript is unique in explaining their in vivo mechanism of action.

Reviewer #1:

*1) This is an interesting study but it almost entirely omits consideration of the critical role of B-cells as APCs in Tfh induction. Previously it has been demonstrated that Tfh form following a two-step process by which T-cells initially interact with DCs which leads to an early wave of pre-Tfh production before a second step of interaction with antigen presenting B-cells that cements the Tfh program (*[9]*). It is surprising that the authors have only focused on the first of these two steps and not examined if there is also any effect on B-cell antigen presentation as this would widen and strengthen the article’s conclusions. An effect on B-cells would not be mutually exclusive to the effects on DCs and it seems possible that both would play a role in the increased Tfh formation seen here. The authors correctly state that the Tangye group (*[11]*) have demonstrated that it is possible to bypass the requirement for B-cells by adding more antigen and prolonging DC antigen presentation, but even here it remains controversial if DCs alone can induce PD1hi GC-Tfh and high levels of IL-21 expression (*[16]*) and where both DCs and B-cells are present B-cells are believed to have a dominant role in Tfh formation from day 3 onwards even in the presence of large amounts of antigen (LCMV infection) (*[7]*).*

*This is important as the authors make clear statements ascribing all the phenomenon to the increased antigen contact in DCs without exploring this alternative possibility. This is particularly acute in*
Figure 5
*where MHCII blocking antibodies are used to interfere with the late stage DC/T-cell contacts. At 72 hours we would also expect B-cell antigen presentation to have an important/dominant role as APCs which would also be blocked by the same antibody, currently it is not possible to say if the effects of the antibody is only by blocking DCs or both DCs and B-cells. It is also worth noting that essentially this point would be testable using the same experimental methods already used i.e. YAe antibody to see if B-cells had altered antigen presentation. Additionally some experiments using vaccination in B-cell deficient mice ideally (μMT, T-cell transfer into RAG-/- or B-cell depletion by anti-CD20) are required to make strong conclusions about the specific role of DCs in the absence of B-cells.*

As the reviewer correctly states, antigen presentation by B cells could be examined using the methods we had employed for the DCs. We had previously generated these data but had found no significant differences in B cell antigen presentation at later times, hence we focused our interest on DCs. We have now included this pMHCII data using the YAe system tracking presentation by B cells following immunization (Figure 5). Also, as noted in the response to Reviewers 2 and 3 below, we have performed the suggested experiments in BcR Tg animals which we feel suffer less drawbacks than the B cell deficient strains suggested. These studies revealed that 200 nm particles were still able to induce a larger population of CXCR5^+^PD-1^+^ cells 5 days post immunization in the absence of cognate interactions with antigen specific B cells, consistent with our other data implicating a role for presentation by DCs.

2) A wider range of sizes (e.g. 40, 100, 150, 200nm) would have been useful to more precisely investigate the relationship between particulate size, DC antigen presentation and effectiveness. For example Mottram et al. used a range of 7 bead sizes and found a bimodal distribution of effectiveness for cytokine induction suggesting that the relationship between size and efficacy may be not always be entirely linear, it would be impossible to make any similar observation in this study since only two sizes are used.

We now include data demonstrating the impact of wider range of particle sizes (20, 40, 100, 200 and 1000nm). Induction of antigen specific IgG2c and IgG1 are shown in Figure 1.

Reviewer #2:

*1) In 2011 in JI Baumjohann et al. described that B cells are dispensable for initial Tfh differentiation (48hrs post vaccination) but played an important role 3.5 days after vaccination for stabilising the Tfh population. There is no discussion in this paper of whether these particles may be acting via Ag-presentation by B cells. Indeed, the evidence for changing the interaction time of T cells and DCs is clear, but the inference that the increased population of Tfh cells observed with large particles is due to these T:DC rather than T:B interactions at 72hrs post immunisation has not been demonstrated. The blocking experiments in*
Figure 5
*would disrupt all antigen presentation, not just that from DCs. A way to determine this experimentally would be to immunise B cell deficient mice (or mice whose B cells don't present cognate antigen) with the different sized particles and see if the expansion of the Tfh population still occurs.*

Our rationale for focusing on DCs at this time point rather than B cells had been due to pMHCII data demonstrating that not only was it prolonged in DCs but that there was no significant difference detected on B cells at this time. Unfortunately, we had not included these data in our original submission, but they are now presented in Figure 5. We agree that this alone does not directly address the contribution of cognate B cell interactions to the effect of 200 nm particles on Tfh differention. As such, we have performed the experiment suggested by the reviewer and compared 200 nm particle induction of Tfh cells in C57BL/6 and MD4 BcR transgenic mice (i.e. in the absence of cognate B cell interactions), and these studies show these cognate interactions were not required. We have now included these data along with reference to the Baumjohann et al. JI paper.

*2) In*
Figure 3
*it is clear that the % of DCs presenting antigen is the same for the first 48hrs. Is the amount (or MFI) of antigen presentation similar between particle sizes, particularly at earlier time points?*

*Benson et al. argue that 200 nm particles prolong the duration of peptide/MHCII presentation on DCs. Does this mean that each individual MHCII molecule is maintained at the cell surface for longer? Or that larger particles deliver processed peptide at a slower rate, perhaps via an alternative route, so that MHCII can be loaded in the ER for longer? Or is it both? Do the authors know how the antigen is processed by the DC when conjugated to 40 nm or 200 nm nanoparticles? Are they processed by different compartments? The stage 3 disruption experiments in*
Figure 5
*define stage 3 simply by time after immunisation; so a slowly loaded MHCII could confound this approach. It would be worth discussing these potential confounders of the interpretation.*

Similar levels of DCs pEaMHCII presentation (by MFI) are seen across the bead sizes during the first 48hrs. The levels detected are comparable with those seen on DCs in the 200 nm particle group at 72 hours (Figure 4). A trend for lower numbers of DCs presenting pEaMHCII in the first 12 hours following immunisation with 200 nm particles was seen, however this was never statistically significant. We agree with the reviewer that the fine detail of why pMHCII is available for longer in the DC compartment is an interesting point of discussion and likely represents a combination of the processes mentioned. We have now included these points in the Discussion and referenced our recent paper investigating these issues for alum adjuvants.

Reviewer #3:

In this manuscript, the authors analyze the relative capacity of particulate antigens to optimize antibody response by inducing dendritic cell-mediated Tfh cell development. They found that:

*1) Ag conjugated 200 nm particles induced more enhanced antibody responses than 40 nm particles or soluble antigen (*Figure 1*). These data are redundant with previous literature showing that nanoparticles allow for the reduction of the administered antigen amount compared to immunization with soluble protein and induce strongly enhanced antibody responses (Virology Journal 2013, 10:108).*

As the reviewer states, particulate antigens have been known to be more effective than soluble in induction of adaptive responses for decades. The data in Figure 1 is included to demonstrate the phenomenon, before we moved on to address the question of why particulate antigens more effectively support an antibody response.

*2) Ag conjugated 200 nm particles induced prolonged presentation of antigen by dendritic cells compared with 40 nm particles or soluble antigen (*Figure 3*). These data are redundant with previous literature showing that particulate form of antigens are more efficiently processed and presented by different APC, including dendritic cells, macrophages and B cells, than soluble antigen (Journal of Immunology, 1996, 156:2809; Immunology and Cell Biology, 2004 82:506).*

While Vidard et al. (JI, 1996) extensively and elegantly demonstrated different capacities for in vitro stimulation of T cell hybridomas by B cell lines and ex-vivo bone marrow differentiated macrophages in response to particle size, the complexity of antigen presentation in vivo setting is not addressed. Thus, by directly immunizing mice with a trackable antigen, we have demonstrated when the processed antigen is presented in vivo and which physiological cell population is doing this. Hence we feel these data are not redundant but is of value to our manuscript and to the field. Additionally, the particulates used throughout our study only differ in size lacking traditional ‘immunostimulatory’ adjuvant properties that would confound analysis of processing, presentation and costimulation (Gamvrellis et al., Immunol Cell Biol, 2004). Thus our data allow direct comparison of in vivo availability of pMHCII in vivo when the only variable is antigen size.

3) Prolonged antigen presentation by dendritic cells drove Tfh cell development. These data are redundant with previous literature showing that dendritic cells support prolonged Tfh cell responses when they are forced to sustained antigen stimulation (Journal of immunology, 2011, 187:1091; Immunity, 2010, 33:241).

The aim of the manuscript was not to decipher key biological events in the induction of Tfh cell differentiation. Rather the aim was to delineate the underlying process by which a single variable of an antigen delivery system (i.e. size) stimulates an adaptive immune response. Thus we believe the novelty lies in the linking of particle size to in vivo presentation kinetics/dynamics, the activation and differentiation of CD4 T cells and induction of protective antibody responses.

Although the authors provide a comprehensive examination of the in vivo T/dendritic cell interactions following in vivo administration of particulate forms of antigens, the lack of conceptual novelty tampers down the enthusiasm for this manuscript. In addition, the authors did not evaluate the contribution of antigen presentation by B cells, and this strongly compromises the conclusions of the paper.

Dendritic cell-mediated presentation launches the Tfh cell program. However, later antigen presentation by B cells is important for the maintenance of Tfh cell responses. Therefore, to demonstrate the ability of the dendritic cells to sustain Tfh cell development, the authors should perform experiments in the absence of B cells or in the absence of key cognate T-B interactions.

To address the contribution of B cells, we have included pMHCII presentation data using the YAe/EαGFP system for this population. Our rationale for focusing on DCs at this time point rather than B cells had been due to pMHCII data demonstrating that not only was it prolonged in DCs but that there was no significant difference detected on B cells at this time. Unfortunately, we had not included these data in our original submission, but they are now presented in Figure 5. However we agree that this alone does not directly address the contribution of cognate B cell interactions to the effect of 200 nm particles on Tfh differention. As such, we have performed the experiment suggested by the reviewer, and compared 200 nm particle induction of Tfh cells in C57BL/6 and MD4 BcR transgenic mice (i.e. in the absence of cognate B cell interactions – Figure 8). Our results demonstrate that 200 nm particles were still able to induce a larger population of CXCR5^+^PD-1^+^ cells 5 days post immunization, consistent with our other data implicating a role for presentation by DCs.